# Noise2Same: Optimizing A Self-Supervised Bound for Image Denoising

**Yaochen Xie**
Texas A&M University
College Station, TX 77843
ethanycx@tamu.edu

**Zhengyang Wang**
Texas A&M University
College Station, TX 77843
zhengyang.wang@tamu.edu

**Shuiwang Ji**
Texas A&M University
College Station, TX 77843
sji@tamu.edu

## Abstract

Self-supervised frameworks that learn denoising models with merely individual noisy images have shown strong capability and promising performance in various image denoising tasks. Existing self-supervised denoising frameworks are mostly built upon the same theoretical foundation, where the denoising models are required to be $\mathcal{J}$-invariant. However, our analyses indicate that the current theory and the $\mathcal{J}$-invariance may lead to denoising models with reduced performance. In this work, we introduce *Noise2Same*, a novel self-supervised denoising framework. In *Noise2Same*, a new self-supervised loss is proposed by deriving a self-supervised upper bound of the typical supervised loss. In particular, *Noise2Same* requires neither $\mathcal{J}$-invariance nor extra information about the noise model and can be used in a wider range of denoising applications. We analyze our proposed *Noise2Same* both theoretically and experimentally. The experimental results show that our *Noise2Same* consistently outperforms previous self-supervised denoising methods in terms of denoising performance and training efficiency.

## 1 Introduction

The quality of deep learning methods for signal reconstruction from noisy images, also known as deep image denoising, has benefited from the advanced neural network architectures such as ResNet [8], U-Net [19] and their variants [29, 16, 26, 31, 25, 14]. While more powerful deep image denoising models are developed over time, the problem of data availability becomes more critical.

Most deep image denoising algorithms are supervised methods that require matched pairs of noisy and clean images for training [27, 29, 2, 7]. The problem of these supervised methods is that, in many denoising applications, the clean images are hard to obtain due to instrument or cost limitations. To overcome this problem, *Noise2Noise* [13] explores an alternative training framework, where pairs of noisy images are used for training. Here, each pair of noisy images should correspond to the same but unknown clean image. Note that *Noise2Noise* is basically still a supervised method, just with noisy supervision.

Despite the success of *Noise2Noise*, its application scenarios are still limited as pairs of noisy images are not available in some cases and may have registration problems. Recently, various of denoising frameworks that can be trained on individual noisy images [23, 17, 28, 10, 1, 12] have been developed. These studies can be divided into two categories according to the amount of extra information required. Methods in the first category requires the noise model to be known. For example, the simulation-based methods [17, 28] use the noise model to generate simulated noises and make individual noisy images noisier. Then a framework similar to *Noise2Noise* can be applied to train the model with pairs of noisier image and the original noisy image. The limitation is obvious as the noise model may be too complicated or even not available.

On the other hand, algorithms in the second category target at more general cases where only individual noisy images are available without any extra information [23, 10, 1, 12]. In this category, self-supervised learning [30, 6, 24] has been widely explored, such as *Noise2Void* [10], *Noise2Self* [1], and the *convolutional blind-spot neural network* [12]. Note that these self-supervised models can be improved as well if information about the noise model is given. For example, Laine et al. [12] and Krull et al. [11] propose the Bayesian post-processing to utilize the noise model. However, with the proposed post-processing, these methods fall into the first category where applicability is limited.

In this work, we stick to the most general cases where only individual noisy images are provided and focus on the self-supervised framework itself without any post-processing step. We note that all of these existing self-supervised denoising frameworks are built upon the same theoretical background, where the denoising models are required to be $\mathcal{J}$-invariant (Section 2). We perform in-depth analyses on the $\mathcal{J}$-invariance property and argue that it may lead to denoising models with reduced performance. Based on this insight, we propose *Noise2Same*, a novel self-supervised denoising framework, with a new theoretical foundation. *Noise2Same* comes with a new self-supervised loss by deriving a self-supervised upper bound of the typical supervised loss. In particular, *Noise2Same* requires neither $\mathcal{J}$-invariance nor extra information about the noise model. We analyze the effect of the new loss theoretically and conduct thorough experiments to evaluate *Noise2Same*. Result show that our *Noise2Same* consistently outperforms previous self-supervised denoising methods.

## 2   Background and Related Studies

**Self-Supervised Denoising with $\mathcal{J}$-Invariant Functions.**   We consider the reconstruction of a noisy image $\boldsymbol{x} \in \mathbb{R}^m$, where $m = (d\times)h \times w \times c$ depends on the spatial and channel dimensions. Let $\boldsymbol{y} \in \mathbb{R}^m$ denotes the clean image. Given noisy and clean image pairs $(\boldsymbol{x}, \boldsymbol{y})$, supervised methods learn a denoising function $f : \mathbb{R}^m \to \mathbb{R}^m$ by minimizing the supervised loss $\mathcal{L}(f) = \mathbb{E}_{x,y} \|f(\boldsymbol{x}) - \boldsymbol{y}\|^2$.

When neither clean images nor paired noisy images are available, various self-supervised denoising methods have been developed [10, 1, 12] by assuming that the noise is zero-mean and independent among all dimensions. These methods are trained on individual noisy images to minimize the self-supervised loss $\mathcal{L}(f) = \mathbb{E}_x \|f(\boldsymbol{x}) - \boldsymbol{x}\|^2$. Particularly, in order to prevent the self-supervised training from collapsing into leaning the identity function, Batson et al. [1] point out that the denoising function $f$ should be $\mathcal{J}$-invariant, as defined below.

**Definition 1.** *For a given partition $\mathcal{J} = \{J_1, \cdots, J_k\}$ ($|J_1| + \cdots + |J_k| = m$) of the dimensions of an image $\boldsymbol{x} \in \mathbb{R}^m$, a function $f : \mathbb{R}^m \to \mathbb{R}^m$ is $\mathcal{J}$-invariant if $f(\boldsymbol{x})_J$ does not depend on $\boldsymbol{x}_J$ for all $J \in \mathcal{J}$, where $f(\boldsymbol{x})_J$ and $\boldsymbol{x}_J$ denotes the values of $f(\boldsymbol{x})$ and $\boldsymbol{x}$ on $J$, respectively.*

Intuitively, $\mathcal{J}$-invariance means that, when denoising $\boldsymbol{x}_J$, $f$ only uses its context $\boldsymbol{x}_{J^c}$, where $J^c$ denotes the complement of $J$. With a $\mathcal{J}$-invariant function $f$, we have

$$\mathbb{E}_x \|f(\boldsymbol{x}) - \boldsymbol{x}\|^2 = \mathbb{E}_{x,y} \|f(\boldsymbol{x}) - \boldsymbol{y}\|^2 + \mathbb{E}_{x,y} \|\boldsymbol{x} - \boldsymbol{y}\|^2 - 2 \langle f(\boldsymbol{x}) - \boldsymbol{y}, \boldsymbol{x} - \boldsymbol{y} \rangle \quad (1)$$

$$= \mathbb{E}_{x,y} \|f(\boldsymbol{x}) - \boldsymbol{y}\|^2 + \mathbb{E}_{x,y} \|\boldsymbol{x} - \boldsymbol{y}\|^2 . \quad (2)$$

Here, the third term in Equation (1) becomes zero when $f$ is $\mathcal{J}$-invariant and the zero-mean assumption about the noise holds [1]. We can see from Equation (2) that when $f$ is $\mathcal{J}$-invariant, minimizing the self-supervised loss $\mathbb{E}_x \|f(\boldsymbol{x}) - \boldsymbol{x}\|^2$ indirectly minimizes the supervised loss $\mathbb{E}_{x,y} \|f(\boldsymbol{x}) - \boldsymbol{y}\|^2$.

All existing self-supervised denoising methods [10, 1, 12] compute the $\mathcal{J}$-invariant denoising function $f$ through a blind-spot network. Concretely, a subset $J$ of the dimensions are sampled from the noisy image $\boldsymbol{x}$ as "blind spots". The blind-spot network $f$ is asked to predict the values of these "blind spots" based on the context $\boldsymbol{x}_{J^c}$. In other words, $f$ is blind on $J$. In previous studies, the blindness on $J$ is achieved in two ways. Specifically, *Noise2Void* [10] and *Noise2Self* [1] use masking, while the *convolutional blind-spot neural network* [12] shifts the receptive field. With the blind-spot network, the self-supervised loss $\mathbb{E}_x \|f(\boldsymbol{x}) - \boldsymbol{x}\|^2$ can be written as

$$\mathcal{L}(f) = \mathbb{E}_J \mathbb{E}_{\boldsymbol{x}} \|f(\boldsymbol{x}_{J^c})_J - \boldsymbol{x}_J\|^2 . \quad (3)$$

While these methods have achieved good performance, our analysis in this work indicates that minimizing the self-supervised loss in Equation (3) with $\mathcal{J}$-invariant $f$ is not optimal for self-supervised denoising. Based on this insight, we propose a novel self-supervised denoising framework,

known as *Noise2Same*. In particular, our *Noise2Same* minimizes a new self-supervised loss without requiring the denoising function $f$ to be $\mathcal{J}$-invariant.

**Bayesian Post-Processing.** From the probabilistic view, the blind-spot network $f$ attempts to model $p(\boldsymbol{y}_J | \boldsymbol{x}_{J^c})$, where the information from $\boldsymbol{x}_J$ is not utilized thus limiting the performance. This limitation can be overcome through the Bayesian deep learning [9] if the noise model $p(\boldsymbol{x}|\boldsymbol{y})$ is known, as proposed by [12, 11]. Specifically, they propose to compute the posterior by

$$p(\boldsymbol{y}_J | \boldsymbol{x}_J, \boldsymbol{x}_{J^c}) \propto p(\boldsymbol{x}_J | \boldsymbol{y}_J) \, p(\boldsymbol{y}_J | \boldsymbol{x}_{J^c}), \; \forall J \in \mathcal{J}. \tag{4}$$

Here, the prior $p(\boldsymbol{y}_J | \boldsymbol{x}_{J^c})$ is Gaussian, whose the mean comes from the original outputs of the blind-spot network $f$ and the variance is estimated by extra outputs added to $f$. The computation of the posterior is a post-processing step, which takes information from $\boldsymbol{x}_J$ into consideration.

Despite the improved performance, the Bayesian post-processing has limited applicability as it requires the noise model $p(\boldsymbol{x}_J | \boldsymbol{y}_J)$ to be knwon. Besides, it assumes that a single type of noise is present for all dimensions. In practice, it is common to have unknown noise models, inconsistent noises, or combined noises with different types, where the Bayesian post-processing is no longer applicable.

In contrast, our proposed *Noise2Same* can make use of the entire input image without any post-processing. Most importantly, *Noise2Same* does not require the noise model to be known and thus can be used in a much wider range of denoising applications.

## 3 Analysis of the $\mathcal{J}$-Invariance Property

In this section, we analyze the $\mathcal{J}$-invariance property and motivate our work. In section 3.1, we experimentally show that the denoising functions trained through mask-based blind-spot methods are not strictly $\mathcal{J}$-invariant. Next, in Section 3.2, we argue that minimizing $\mathbb{E}_x \|f(\boldsymbol{x}) - \boldsymbol{x}\|^2$ with $\mathcal{J}$-invariant $f$ is not optimal for self-supervised denoising.

### 3.1 Mask-Based Blind-Spot Denoising: Is the Optimal Function Strictly $\mathcal{J}$-Invariant?

We show that, in mask-based blind-spot approaches, the optimal denoising function obtained through training is not strictly $\mathcal{J}$-invariant, which contradicts the theory behind these methods. As introduced in Section 2, mask-based blind-spot methods implement blindness on $J$ through masking. Original values on $J$ are masked out and replaced by other values. Concretely, in Equation (3), $\boldsymbol{x}_{J^c}$ becomes $\mathbb{1}_{J^c} \cdot \boldsymbol{x} + \mathbb{1}_J \cdot \boldsymbol{r}$, where $\boldsymbol{r}$ denotes the new values on the masked locations $(J)$. As introduced in Section 2, *Noise2Void* [10] and *Noise2Self* [1] are current mask-based blind-spot methods. The main difference between them is the choice of the replacement strategy, *i.e.*, how to select $\boldsymbol{r}$. Specifically, *Noise2Void* applies the Uniform Pixel Selection (UPS) to randomly select $\boldsymbol{r}$ from local neighbors of the masked locations, while *Noise2Self* directly uses a random value.

Although the masking prevents $f$ from accessing the original values on $J$ during training, we point out that, during inference, $f$ still shows a weak dependency on values on $J$, and thus does not strictly satisfy the $\mathcal{J}$-invariance property. In other words, mask-based blind-spot methods do not guarantee the learning of a $\mathcal{J}$-invariant function $f$. We conduct experiments to verify the above statement. Concretely, given a denoising function $f$ trained through mask-based blind-spot methods, we quantify the strictness of $\mathcal{J}$-invariance by computing the following metric:

$$\mathcal{D}(f) = \mathbb{E}_J \mathbb{E}_x \|f(\boldsymbol{x}_{J^c})_J - f(\boldsymbol{x})_J\|^2 / |J|, \tag{5}$$

where $x$ is the raw noisy image and $\boldsymbol{x}_{J^c}$ denotes the image whose values on $J$ are replaced with random Gaussian noises ($\sigma_m$=0.5). Note that the replacement here is irrelevant to the the replacement strategy used in mask-based blind-spot methods. If the function $f$ is strictly $\mathcal{J}$-invariant, $\mathcal{D}(f)$ should be close to 0 for all $\boldsymbol{x}$. Smaller $\mathcal{D}(f)$ indicates more $\mathcal{J}$-invariant $f$. To mitigate mutual influences among the locations within $J$, we use saturate sampling [10] to sample $J$ and make the sampling sparse enough (at a portion of 0.01%). $\mathcal{D}(f)$ is computed on the output of $f$ before re-scaling back to [0,255]. In our experiments, we compare $\mathcal{D}(f)$ and the testing PSNR for $f$ trained with different replacement strategies and on different datasets.

Table 1: $\mathcal{D}(f)$ and PSNR of $f$ trained through mask-based blind-spot methods with different replacement strategies on BSD68. The last column corresponds to a strictly $\mathcal{J}$-invariant model.

| Replacement Strategy | Gaussian ($\sigma$=0.2) | Gaussian ($\sigma$=0.5) | Gaussian ($\sigma$=0.8) | Gaussian ($\sigma$=1.0) | UPS ($5 \times 5$) | Shifting RF |
|---|---|---|---|---|---|---|
| $\mathcal{D}(f)$ ($\times 10^{-3}$) | 4.326 | 10.91 | 2.141 | 1.569 | 18.31 | 0.105 |
| PSNR | 26.14 | 26.83 | 26.85 | 26.98 | 27.71 | 27.15 |

Table 1 provides the comparison results between $f$ trained with different replacement strategies on the BSD68 dataset [15]. We also include the scores of the *convolutional blind-spot neural network* [12] for reference, which guarantees the strict $\mathcal{J}$-invariance through shifting receptive field, as discussed in Section 3.2. As expected, it has a close-to-zero $\mathcal{D}(f)$, where the non-zero value comes from mutual influences among the locations within $J$ and the numerical precision. The large $\mathcal{D}(f)$ for all the mask-based blind-spot methods indicate that the $\mathcal{J}$-invariance is not strictly guaranteed and the strictness varies significantly over different replacement strategies.

Table 2: $\mathcal{D}(f)$ and PSNR of $f$ on trained through mask-based blind-spot methods with the same replacement strategy on different datasets.

| Datasets | BSD68 | HanZi | ImageNet |
|---|---|---|---|
| $\mathcal{D}(f)$ ($\times 10^{-3}$) | 10.91 | 0.249 | 17.67 |
| PSNR | 26.83 | 13.94 | 20.38 |

We also compare results on different datasets when we fix the replacement strategy, as shown in Table 2. We can see that different datasets have strong influences on the strictness of $\mathcal{J}$-invariance as well. Note that such influences are not under the control of the denoising approach itself. In addition, although the shown results in Tables 1 and 2 are computed on testing dataset at the end of training, similar trends with $\mathcal{D}(f) \gg 0$ is observed during training.

Given the results in Tables 1 and 2, we draw our conclusions from two aspects. We first consider the mask together with the network $f$ as a $\mathcal{J}$-invariant function $g$, *i.e.*, $g(x) := f(\mathbb{1}_{J^c} \cdot \boldsymbol{x} + \mathbb{1}_J \cdot \boldsymbol{r})$. In this case, the function $g$ is guaranteed to be $\mathcal{J}$-invariant during training, and thus Equation (2) is valid. However, during testing, the mask is removed and a different non-$\mathcal{J}$-invariant function $f$ is used because $f$ achieves better performance than $g$, according to [1]. This contradicts the theoretical results of [1]. On the other hand, we consider the network $f$ and the mask separately and perform training and testing with the same function $f$. In this case, the use of mask aims to help $f$ learn to be $\mathcal{J}$-invariant during training so that Equation (2) becomes valid. However, our experiments show that $f$ is neither strictly $\mathcal{J}$-invariant during training nor till the end of training, indicating that Equation (2) is not valid. With findings interpreted from both aspects, we ask whether minimizing $\mathbb{E}_x \left\| f(\boldsymbol{x}) - \boldsymbol{x} \right\|^2$ with $\mathcal{J}$-invariant $f$ yields optimal performance for self-supervised denoising.

## 3.2 Shifting Receptive Field: How do the Strictly $\mathcal{J}$-Invariant Models Perform?

We directly show that, with a strictly $\mathcal{J}$-invariant $f$, minimizing $\mathbb{E}_x \left\| f(\boldsymbol{x}) - \boldsymbol{x} \right\|^2$ does not necessarily lead to the best performance. Different from mask-based blind-spot methods, Laine et al. [12] propose the *convolutional blind-spot neural network*, which achieves the blindness on $J$ by shifting receptive field (RF). Specifically, each pixel in the output image excludes its corresponding pixel in the input image from its receptive field. As values outside the receptive field cannot affect the output, the *convolutional blind-spot neural network* is strictly $\mathcal{J}$-invariant by design.

According to Table 1, the shift RF method outperforms all the mask-based blind-spot approaches with Gaussian replacement strategies, indicating the advantage of the strict $\mathcal{J}$-invariance. However, we notice that the UPS replacement strategy shows a different result. Here, a denoising function with less strict $\mathcal{J}$-invariance performs the best. One possible explanation is that the UPS replacement has a certain probability to replace a masked location by its original value. It weakens the $\mathcal{J}$-invariance of the mask-based denoising model but boosts the performance by yielding a result that is equivalent to computing a linear combination of the noisy input and the output of a strictly $\mathcal{J}$-invariant blind-spot network [1]. This result shows that minimizing $\mathbb{E}_x \left\| f(\boldsymbol{x}) - \boldsymbol{x} \right\|^2$ with a strictly $\mathcal{J}$-invariant $f$ does not necessarily give the best performance. Another evidence is the Bayesian post-processing introduced in Section 2, which also make the final denoising function not strictly $\mathcal{J}$-invariant while boosting the performance.

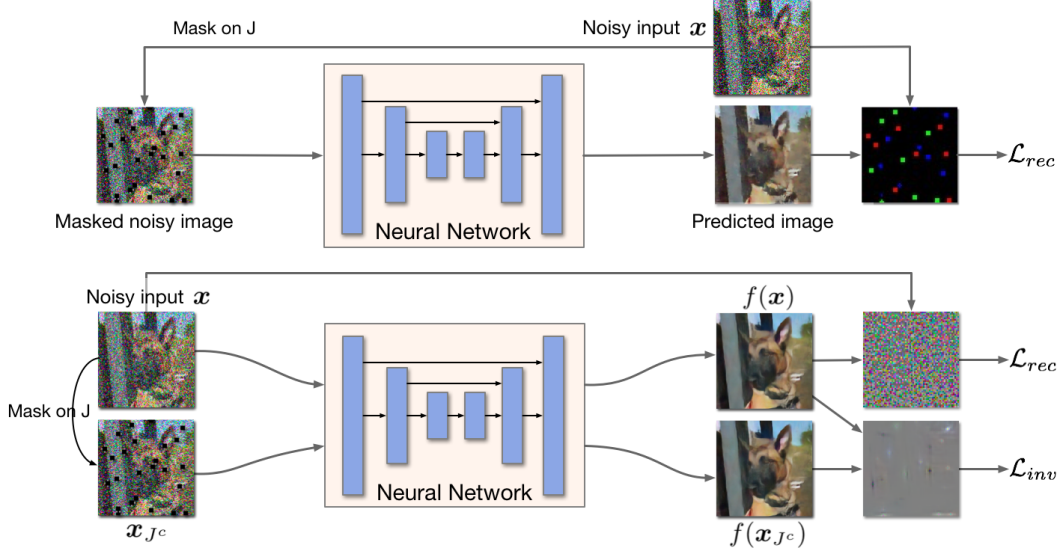

Figure 1: **Top**: The framework of the mask-based blind-spot denoising methods. The neural network takes the masked noisy image and predicts the masked value. The reconstruction loss is only computed on the masked dimensions. **Bottom**: The *Noise2Same* framework. The neural network takes both the full noisy image and the masked image as inputs and produces two outputs. The reconstruction loss is computed between the full noisy image and its corresponding output. The invariance loss is computed between the two outputs.

To conclude, we argue that minimizing $\mathbb{E}_x \left\| f(\boldsymbol{x}) - \boldsymbol{x} \right\|^2$ with $\mathcal{J}$-invariant $f$ can lead to reduction in performance for self-supervised denoising. In this work, we propose a new self-supervised loss. Our loss does not require the $\mathcal{J}$-invariance. In addition, our proposed method can take advantage of the information from the entire noisy input without any post-processing step or extra assumption about the noise.

## 4 The Proposed Noise2Same Method

In this section, we introduce *Noise2Same*, a novel self-supervised denoising framework. *Noise2Same* comes with a new self-supervised loss. In particular, *Noise2Same* requires neither $\mathcal{J}$-invariant denoising functions nor the noise models.

### 4.1 Noise2Same: A Self-Supervised Upper Bound without the $\mathcal{J}$-Invariance Requirement

As introduced in Section 2, the $\mathcal{J}$-invariance requirement sets the inner product term $\langle f(\boldsymbol{x}) - \boldsymbol{y}, \boldsymbol{x} - \boldsymbol{y} \rangle$ in Equation (1) to zero. The resulting Equation (2) shows that minimizing $\mathbb{E}_x \left\| f(\boldsymbol{x}) - \boldsymbol{x} \right\|^2$ with $\mathcal{J}$-invariant $f$ indirectly minimizes the supervised loss, leading to the current self-supervised denoising framework. However, we have pointed out that this framework yields reduced performance.

In order to overcome this limitation, we propose to control the right side of Equation (2) with a self-supervised upper bound, instead of approximating $\langle f(\boldsymbol{x}) - \boldsymbol{y}, \boldsymbol{x} - \boldsymbol{y} \rangle$ to zero. The upper bound holds without requiring the denoising function $f$ to be $\mathcal{J}$-invariant.

**Theorem 1.** *Consider a normalized noisy image $\boldsymbol{x} \in \mathbb{R}^m$ (obtained by subtracting the mean and dividing by the standard deviation) and its ground truth signal $\boldsymbol{y} \in \mathbb{R}^m$. Assume the noise is zero-mean and i.i.d among all the dimensions, and let $J$ be a subset of $m$ dimensions uniformly sampled from the image $\boldsymbol{x}$. For any $f : \mathbb{R}^m \to \mathbb{R}^m$, we have*

$$\mathbb{E}_{x,y} \left\| f(\boldsymbol{x}) - \boldsymbol{y} \right\|^2 + \left\| \boldsymbol{x} - \boldsymbol{y} \right\|^2 \leq \mathbb{E}_x \left\| f(\boldsymbol{x}) - \boldsymbol{x} \right\|^2 + 2m \, \mathbb{E}_J \left[ \frac{\mathbb{E}_x \left\| f(\boldsymbol{x})_J - f(\boldsymbol{x}_{J^c})_J \right\|^2}{|J|} \right]^{1/2} \quad (6)$$

The proof of Theorem 1 is provided in Appendix A. With Theorem 1, we can perform self-supervised denoising by minimizing the right side of Inequality (6) instead. Following the theoretical result,

we propose our new self-supervised denoising framework, *Noise2Same*, with the following self-supervised loss:

$$\mathcal{L}(f) = \mathbb{E}_x \|f(\boldsymbol{x}) - \boldsymbol{x}\|^2 / m + \lambda_{inv} \, \mathbb{E}_J \left[ \mathbb{E}_x \|f(\boldsymbol{x})_J - f(\boldsymbol{x}_{J^c})_J\|^2 / |J| \right]^{1/2}. \tag{7}$$

This new self-supervised loss consists of two terms: a reconstruction mean squared error (MSE) $\mathcal{L}_{rec} = \mathbb{E}_x \|f(\boldsymbol{x}) - \boldsymbol{x}\|^2$ and a squared-root of invariance MSE $\mathcal{L}_{inv} = \mathbb{E}_J(\mathbb{E}_x \|f(\boldsymbol{x})_J - f(\boldsymbol{x}_{J^c})_J\|^2 / |J|)^{1/2}$. Intuitively, $\mathcal{L}_{inv}$ prevents our model from learning the identity function when minimizing $\mathcal{L}_{rec}$ without any requirement on $f$. In fact, by comparing $\mathcal{L}_{inv}$ with $\mathcal{D}(f)$ in Equation (5), we can see that $\mathcal{L}_{inv}$ implicitly controls how strictly $f$ should be $\mathcal{J}$-invariant, avoiding the explicit $\mathcal{J}$-invariance requirement. We balance $\mathcal{L}_{rec}$ and $\mathcal{L}_{inv}$ with a positive scalar weight $\lambda_{inv}$. By default, we set $\lambda_{inv} = 2$ according to Theorem 1. In some cases, setting $\lambda_{inv}$ to different values according to the scale of observed $\mathcal{L}_{inv}$ during training could help achieve a better denoising performance.

Figure 1 compares our proposed *Noise2Same* with mask-based blind-spot denoising methods. Mask-based blind-spot denoising methods employ the self-supervised loss in Equation (3), where the reconstruction MSE $\mathcal{L}_{rec}$ is computed only on $J$. In contrast, our proposed *Noise2Same* computes $\mathcal{L}_{rec}$ between the entire noisy image $\boldsymbol{x}$ and the output of the neural network $f(\boldsymbol{x})$. To compute the invariance term $\mathcal{L}_{inv}$, we still feed the masked noisy image $\boldsymbol{x}_{J^c}$ to the neural network and compute MSE between $f(\boldsymbol{x})$ and $f(\boldsymbol{x}_{J^c})$ on $J$, *i.e.*, $f(\boldsymbol{x})_J$ and $f(\boldsymbol{x}_{J^c})_J$. Note that, while *Noise2Same* also samples $J$ from $\boldsymbol{x}$, it does not require $f$ to be $\mathcal{J}$-invariant.

### 4.2 Analysis of the Invariance Term

The invariance term $\mathcal{L}_{inv}$ is a unique and important part in our proposed self-supervised loss. In this section, we further analyze the effect of this term. To make the analysis concrete, we perform analysis based on an example case, where the noise model is given as the additive Gaussian noise $N(0, \sigma)$. Note that the example is for analysis purpose only, and the application of our proposed *Noise2Same* does not require the noise model to be known.

**Theorem 2.** *Consider a noisy image $\boldsymbol{x} \in \mathbb{R}^m$ and its ground truth signal $\boldsymbol{y} \in \mathbb{R}^m$. Assume the noise is i.i.d among all the dimensions, and let $J$ be a subset of $m$ dimensions uniformly sampled from the image $\boldsymbol{x}$. If the noise is additive Gaussian with zero-mean and standard deviation $\sigma$, we have*

$$\mathbb{E}_{x,y} \|f(\boldsymbol{x}) - \boldsymbol{y}\|^2 + \|\boldsymbol{x} - \boldsymbol{y}\|^2 \leq \mathbb{E}_x \|f(\boldsymbol{x}) - \boldsymbol{x}\|^2 + 2m\sigma \, \mathbb{E}_J \left[ \frac{\mathbb{E} \|f(\boldsymbol{x})_J - f(\boldsymbol{x}_{J^c})_J\|^2}{|J|} \right]^{1/2} \tag{8}$$

The proof of Theorem 2 is provided in Appendix B. Note that the noisy image $\boldsymbol{x}$ here **does not require normalization** as in Theorem 1. Compared to Theorem 1, the $\sigma$ from the noise model is added to balance the invariance term. As introduced in Section 4.1, the invariance term controls how strictly $f$ should be $\mathcal{J}$-invariant and a higher weight of the invariance term pushes the model to learn a more strictly $\mathcal{J}$-invariant $f$. Therefore, Theorem 2 indicates that, when the noise is stronger with a larger $\sigma$, $f$ should be more strictly $\mathcal{J}$-invariant. Based on the definition of $\mathcal{J}$-invariance, a more strictly $\mathcal{J}$-invariant $f$ will depend more on the context $\boldsymbol{x}_{J^c}$ and less on the noisy input $\boldsymbol{x}_J$.

This result is consistent with the findings in previous studies. Batson et al. [1] propose to compute the linear combination of the noisy image and the output of the blind-spot network as a post-processing step, leading to better performance. The weights in the linear combination are determined by the variance of noise. And a higher weight is given to the output of the blind-spot network with larger noise variance. Laine et al. [12] derive a similar result through the Bayesian post-processing. This explains how the invariance term in our proposed *Noise2Same* improves denoising performance.

However, a critical difference between our *Noise2Same* and previous studies is that, the post-processing in [1, 12] cannot be performed when the noise model is unknown. To the contrary, *Noise2Same* is able to control how strictly $f$ should be $\mathcal{J}$-invariant through the invariance term without any assumption about the noise or requirement on $f$. This property allows *Noise2Same* to be used in a much wider range of denoising tasks with unknown noise models, inconsistent noise, or combined noises with different types.

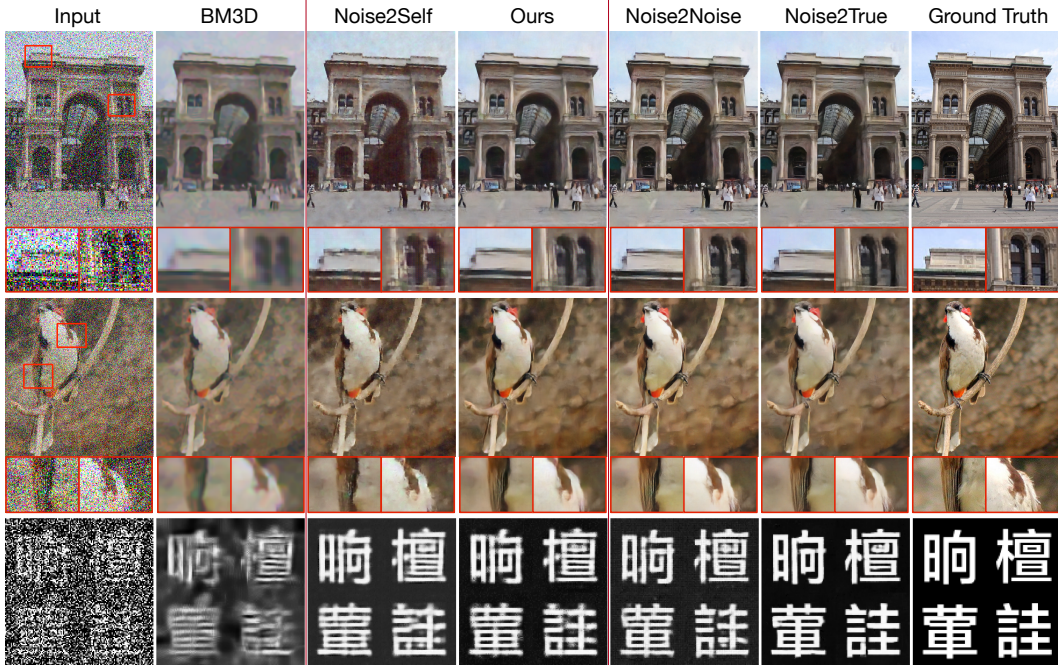

Figure 2: **RGB natural images and hand-written Chinese character images**: Visualizations of testing results on ImageNet dataset (first two rows) and the HànZì Dataset (the third row). We compare the denoising quality among the traditional method *BM3D*, supervised methods *Noise2True* and *Noise2Noise*, self-supervised approaches *Noise2Self* and our *Noise2Same*. From the left to the right, the columns are in the ascending order in terms of the denoising quality.

## 5 Experiments

We evaluate our *Noise2Same* on four datasets, including RGB natural images (ImageNet ILSVRC 2012 Val [21]), generated hand-written Chinese character images (HànZì [1]), physically captured 3D microscopy data (Planaria [27]) and grey-scale natural images (BSD68 [15]). The four datasets have different noise types and levels. The constructions of the four datasets are described in Appendix C.

### 5.1 Comparisons with Baselines

The baselines include traditional denoising algorithms (*NLM* [3], *BM3D* [5]), supervised methods (*Noise2True*, *Noise2Noise* [13]), and previous self-supervised methods (*Noise2Void* [10], *Noise2Self* [1], the *convolutional blind-spot neural network* [12]). Note that we consider *Noise2Noise* as a supervised model since it requires pairs of noisy images, where the supervision is noisy. While *Noise2Void* and *Noise2Self* are similar methods following the blind-spot approach, they mainly differ in the strategy of mask replacement. To be more specific, *Noise2Void* proposes to use Uniform Pixel Selection (UPS), and *Noise2Self* proposes to exclude the information of the masked pixel and uses a random value on the range of given image data. As an additional mask strategy using the local average excluding the center pixel (donut) is mentioned in [1], we also include it for comparison. We use the same neural network architecture for all deep learning methods. Detailed experimental settings are provided in Appendices D and E.

Note that ImageNet and HànZì have combined noises and Planaria has unknown noise models. As a result, the post-processing steps in *Noise2Self* [1] and the *convolutional blind-spot neural network* [12] are not applicable, as explained in Section 2. In order to make fair comparisons under the self-supervised category, we train and evaluate all models only using the images, without extra information about the noise. In this case, among self-supervised methods, only our *Noise2Same* and *Noise2Void* with the UPS replacement strategy can make use of information from the entire input image, as demonstrated in Section 3.2. We also include the complete version of the *convolutional*

Table 3: Comparisons among denoising methods on different datasets, in terms of Peak Signal-to-Noise Ratio (PSNR). The post-processing of Laine et al. [12] that requires information about the noise model is included under the *Self-Supervised + noise model* category and is excluded under the *Self-Supervised* category. Noise2Self-Noise and Noise2Self-Donut refer to two masking strategies mentioned in [1], where the original results presented in [1] are produced by the noise masking. Bold numbers indicate the best performance among self-supervised methods.

| | | Datasets | | | |
|---|---|---|---|---|---|
| | **Methods** | ImageNet | HànZì | Planaria | BSD68 |
| *Traditional* | Input | 9.69 | 6.45 | 21.52 / 21.09 / 20.82 | 20.19 |
| | NLM [3] | 18.04 | 8.41 | 25.80 / 24.03 / 21.62 | 22.73 |
| | BM3D [5] | 18.74 | 10.90 | - | 28.59 |
| *Supervised* | Noise2True | 23.39 | 15.66 | 31.57 / 30.15 / 28.13 | 29.06 |
| | Noise2Noise [13] | 23.27 | 14.30 | - | 28.86 |
| *Self-Supervised + noise model* | Laine et al. [12] | - | - | - | 28.84 |
| *Self-Supervised* | Laine et al. [12] | 20.89 | 10.70 | - | 27.15 |
| | Noise2Void [10] | 21.36 | 13.72 | 25.84 / 23.57 / 21.60 | 27.71 |
| | Noise2Self-Noise [1] | 20.38 | 13.94 | 27.58 / 24.83 / 21.83 | 26.98 |
| | Noise2Self-Donut [1] | 8.62 | 13.29 | 27.63 / 24.72 / 21.73 | **28.20** |
| | **Noise2Same** | **22.26** | **14.38** | **29.48 / 26.93 / 22.41** | 27.95 |

*blind-spot neural network* with post-processing, who is only available on *BSD68*, where the noise is not combined and the noise type is known.

Following previous studies, we use Peak Signal-to-Noise Ratio (PSNR) as the evaluation metric. The comparison results between our *Noise2Same* and the baselines in terms of PSNR on the four datasets are summarized in Table 3 and visualized in Figure 2 and Appendix F. The results show that our *Noise2Same* achieve remarkable improvements over previous self-supervised baselines on ImageNet, HànZì and CARE. In particular, on the ImageNet and the HànZì Datasets, our *Noise2Same* and *Noise2Void* demonstrate the advantage of utilizing information from the entire input image. Although the using of donut masking can achieve better performance on the

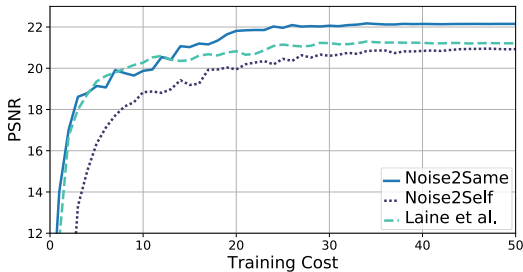

Figure 3: **Training efficiency**. For a fair comparison, we adjust the batch sizes for each method to fill the memory of a single GPU, namely, 128 for *Noise2Self*, 64 for *Noise2Same* and 32 for *Laine et al.* One unit of training cost represents 50 minibatch steps.

BSD68 Dataset, it leads to model collapsing on the ImageNet Dataset and hence can be unstable. On the other hand, the *convolutional blind-spot neural network* [12] suffers from significant performance losses without the Bayesian post-processing, which requires information about the noise models that are unknown.

We note that, in our fair settings, supervised methods still have better performance over self-supervised models, especially on the Planaria and BSD68 datasets. One explanation is that the supervision usually carries extra information implicitly, such as information about the noise model. Here, we draw a conclusion different from Batson et al. [1]. That is, there are still performance gaps between self-supervised and supervised denoising methods. Our *Noise2Same* moves one step towards closing the gap by proposing a new self-supervised denoising framework.

In addition to the performance, we compares the training efficiency among self-supervised methods as well. Specifically, we plot how the PSNR changes during training on the ImageNet dataset. We compare *Noise2Same* with *Noise2Self* and the *convolutional blind-spot neural network*. The plot shows that our *Noise2Same* has similar convergence speed to the *convolutional blind-spot neural*

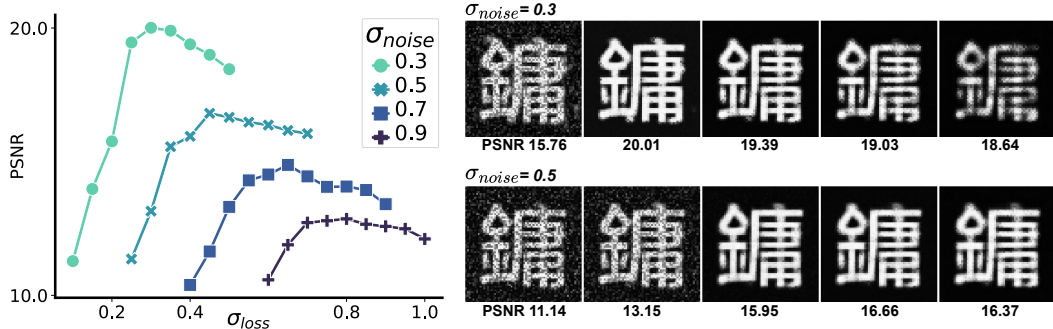

Figure 4: **Effect of the invariance term**. **Left**: Given additive Gaussian noise with certain $\sigma_{noise}$, how the performance of our *Noise2Same* varies over different $\sigma_{loss}$. **Right**: We visualize some denoising examples from noisy images with $\sigma_{noise} = 0.3, 0.5$. From left to right, the columns correspond to setting $\sigma_{loss}$ to $0.2, 0.3, 0.4, 0.5, 0.6$, respectively.

*network*. On the other hand, as the mask-based method *Noise2Self* uses only a subset of output pixels to compute the loss function in each step, the training is expected to be slower [12].

## 5.2 Effect of the Invariance Term

In Section 4.2, we analyzed the effect of the invariance term using an example, where the noise model is given as the additive Gaussian noise. In this example, the variance of the noise controls how the strictness of the optimal $f$ through the coefficient $\lambda_{inv}$ of the invariance term.

Here, we conduct experiments to verify this insight. Specifically, we construct four noisy dataset from the HànZì dataset with only additive Gaussian noise at different levels ($\sigma_{noise} = 0.9, 0.7, 0.5, 0.3$). Then we train *Noise2Same* with $\lambda_{inv} = 2\sigma_{loss}$ by varying $\sigma_{loss}$ from $0.1$ to $1.0$ for each dataset. According to Theorem 2, the best performance on each dataset should be achieved when $\sigma_{loss}$ is close to $\sigma_{noise}$. The results, as reported Figure 4, are consistent with our theoretical results in Theorem 2.

## 6 Conclusion and Future Work

We analyzed the existing blind-spot-based denoising methods and introduced *Noise2Same*, a novel self-supervised denoising method, which removes the assumption and over-restriction on the neural network as a $\mathcal{J}$-invariant function. We provided further analysis on *Noise2Same* and experimentally demonstrated the denoising capability of *Noise2Same*. As an orthogonal work, the combination of self-supervised denoising result and the noise model has be shown to provide additional performance gain. We would like to further explore noise model-augmented *Noise2Same* in future works.

## Broader Impact

In this paper, we introduce Noise2Same, a self-supervised framework for deep image denoising. As Noise2Same does not need paired clean data, paired noisy data, nor the noise model, its application scenarios could be much broader than both traditional supervised and existing self-supervised denoising frameworks. The most direct application of Noise2Same is to perform denoising on digital images captured under poor conditions. Individuals and corporations related to photography may benefit from our work. Besides, Noise2Same could be applied as a pre-processing step for computer vision tasks such as object detection and segmentation [18], making the downstream algorithms more robust to noisy images. Also, specific research communities could benefit from the development of Noise2Same as well. For example, the capture of high-quality microscopy data of live cells, tissue, or nanomaterials is expensive in terms of budget and time [27]. Proper denoising algorithms allow researchers to obtain high-quality data from low-quality data and hence remove the need to capture high-quality data directly. In addition to image denoising applications, the self-supervised denoising framework could be extended to other domains such as audio noise reduction and single-cell [1]. On the negative aspect, as many imaging-based research tasks and computer vision applications may be

built upon the denoising algorithms, the failure of Noise2Same could potentially lead to biases or failures in these tasks and applications.

## Acknowledgments and Disclosure of Funding

This work was supported in part by National Science Foundation grant DBI-2028361.

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
