[Supplementary Material]

# A Proof of Theorem 1

*Proof.* We consider the third term on the right-hand side of Equation (1). Instead of reducing the third term $2 \langle f(\boldsymbol{x}) - \boldsymbol{y}, \boldsymbol{x} - \boldsymbol{y} \rangle$ to 0 under the $\mathcal{J}$-invariant assumption, we control this term with its upper bound with the only assumption that $E[\boldsymbol{x}|\boldsymbol{y}] = \boldsymbol{y}$. Formally, we have

$$\mathbb{E}_{x,y} \langle f(\boldsymbol{x}) - \boldsymbol{y}, \boldsymbol{x} - \boldsymbol{y} \rangle = \mathbb{E}_y \mathbb{E}_{x|y} \sum_j (f(\boldsymbol{x})_j - y_j)(x_j - y_j) \tag{9}$$

$$= \sum_j \mathbb{E}_y \left[ \mathbb{E}_{x|y}(f(\boldsymbol{x})_j - y_j)(x_j - y_j) - \mathbb{E}_{x|y}(f(\boldsymbol{x})_j - y_j)\mathbb{E}_{x|y}(x_j - y_j) \right] \tag{10}$$

$$= \sum_j \mathbb{E}_y \left[ \mathrm{Cov}(f(\boldsymbol{x})_j - y_j, x_j - y_j | \boldsymbol{y}) \right] \tag{11}$$

$$= \sum_j \mathbb{E}_y \left[ \mathrm{Cov}(f(\boldsymbol{x})_j, x_j | \boldsymbol{y}) \right]. \tag{12}$$

Equation (10) holds due to the zero-mean assumption, where $\mathbb{E}_{x|y}(x_j - y_j) = 0$. Now we let $J$ be a uniformly sampled subset of the image dimensions $\{1, \cdots, m\}$, then we have the equation

$$\sum_j \mathbb{E}_y \left[ \mathrm{Cov}(f(\boldsymbol{x})_j, x_j | \boldsymbol{y}) \right] = \frac{m}{|J|} \mathbb{E}_J \sum_{j \in J} \mathbb{E}_y \left[ \mathrm{Cov}(f(\boldsymbol{x})_j, x_j | \boldsymbol{y}) \right]. \tag{13}$$

The right-hand side of the equation above can be controlled by applying Cauchy-Schwarz inequality while the input images are normalized. We have, for all $J$,

$$\frac{1}{|J|} \sum_{j \in J} \mathbb{E}_y \left[ \mathrm{Cov}(f(\boldsymbol{x})_j, x_j | \boldsymbol{y}) \right] = \frac{1}{|J|} \sum_{j \in J} \mathbb{E}_y \left[ \mathrm{Cov}(f(\boldsymbol{x})_j - f(\boldsymbol{x}_{J^c})_j, x_j | \boldsymbol{y}) \right] \tag{14}$$

$$\leq \frac{1}{|J|} \sum_{j \in J} \mathbb{E}_y \left[ \mathrm{Var}(f(\boldsymbol{x})_j - f(\boldsymbol{x}_{J^c})_j | \boldsymbol{y}) \cdot \mathrm{Var}(x_j | \boldsymbol{y}) \right]^{1/2} \tag{15}$$

$$\leq \left( \frac{1}{|J|} \sum_{j \in J} \mathbb{E}_y \left[ \mathrm{Var}(f(\boldsymbol{x})_j - f(\boldsymbol{x}_{J^c})_j | \boldsymbol{y}) \cdot \mathrm{Var}(x_j | \boldsymbol{y}) \right] \right)^{1/2} \tag{16}$$

$$\leq \left( \frac{1}{|J|} \sum_{j \in J} \mathbb{E}_y \left[ \mathbb{E}\left[ [f(\boldsymbol{x})_j - f(\boldsymbol{x}_{J^c})_j]^2 \right] | \boldsymbol{y} \right] \right)^{1/2} \tag{17}$$

$$= \left( \frac{1}{|J|} \sum_{j \in J} \mathbb{E}\left[ f(\boldsymbol{x})_j - f(\boldsymbol{x}_{J^c})_j \right]^2 \right)^{1/2} \tag{18}$$

$$= \left( \frac{1}{|J|} \mathbb{E} \left\| f(\boldsymbol{x})_J - f(\boldsymbol{x}_{J^c})_J \right\|^2 \right)^{1/2}. \tag{19}$$

To be more specific, Equation (14) follows since $f(\boldsymbol{x}_{J^c})_J$ does not correlate to $\boldsymbol{x}_j$ due to the independent noise assumption and $j \notin J^c$, and subtracting $f(\boldsymbol{x}_{J^c})_j$ from $f(\boldsymbol{x})_j$ does not change the Covariance. Inequality (15) applies the Cauchy-Schwarz inequality. Inequality (16) holds due to $(\mathbb{E}X)^2 \leq \mathbb{E}X^2$. The derivation of Inequality (17) uses the fact that $\mathrm{Var}(x_j) = 1$ under normalization and $\mathrm{Var}(x_j | \boldsymbol{y}) \leq \mathrm{Var}(x_j) = 1$ for all $j$.

Consequently, we can control Equation (1) as

$$\mathbb{E}_{x,y}\left\|f(\boldsymbol{x})-\boldsymbol{y}\right\|^2 + \mathbb{E}_{x,y}\left\|\boldsymbol{x}-\boldsymbol{y}\right\|^2 = \mathbb{E}_x\left\|f(\boldsymbol{x})-\boldsymbol{x}\right\|^2 + 2\,\mathbb{E}_{x,y}\langle f(\boldsymbol{x})-\boldsymbol{y},\boldsymbol{x}-\boldsymbol{y}\rangle \tag{20}$$

$$\leq \mathbb{E}_x\left\|f(\boldsymbol{x})-\boldsymbol{x}\right\|^2 + 2m\,\mathbb{E}_J\left[\frac{1}{|J|}\mathbb{E}\left\|f(\boldsymbol{x})_J - f(\boldsymbol{x}_{J^c})_J\right\|^2\right]^{1/2}. \tag{21}$$

This completes the proof of Theorem 1.

$\square$

## B  Proof of Theorem 2

*Proof.* We start from Equation (13) in the proof of Theorem 1. Since we have a stronger assumption that the noise model is known to be additive with standard deviation $\sigma$ and zero-mean, we have $\mathrm{Var}(\boldsymbol{x}_j - \boldsymbol{y}_j) = \sigma^2$ for all $j$. Due to that the additive noise is orthogonal to the signal $\boldsymbol{y}$, we futher have the conditional variance $\mathrm{Var}(\boldsymbol{x}_j - \boldsymbol{y}_j|\boldsymbol{y}) = \sigma^2$. Then, similar to the proof of Theorem 1, we have,

$$\frac{1}{|J|}\sum_{j\in J}\mathbb{E}_y\Big[\mathrm{Cov}(f(\boldsymbol{x})_j, x_j|\boldsymbol{y})\Big] = \frac{1}{|J|}\sum_{j\in J}\mathbb{E}_y\Big[\mathrm{Cov}(f(\boldsymbol{x})_j - f(\boldsymbol{x}_{J^c})_j, x_j - y_j|\boldsymbol{y})\Big] \tag{22}$$

$$\leq \frac{1}{|J|}\sum_{j\in J}\mathbb{E}_y\Big[\mathrm{Var}(f(\boldsymbol{x})_j - f(\boldsymbol{x}_{J^c})_j|\boldsymbol{y})\cdot\mathrm{Var}(x_j - y_j|\boldsymbol{y})\Big]^{1/2} \tag{23}$$

$$\leq \left(\frac{1}{|J|}\sum_{j\in J}\mathbb{E}_y\Big[\mathrm{Var}(f(\boldsymbol{x})_j - f(\boldsymbol{x}_{J^c})_j|\boldsymbol{y})\cdot\mathrm{Var}(x_j - y_j|\boldsymbol{y})\Big]\right)^{1/2} \tag{24}$$

$$= \left(\frac{1}{|J|}\sum_{j\in J}\mathbb{E}_y\Big[\mathbb{E}\big[[f(\boldsymbol{x})_j - f(\boldsymbol{x}_{J^c})_j]^2|\boldsymbol{y}\big]\cdot\sigma^2\Big]\right)^{1/2} \tag{25}$$

$$= \sigma\left(\frac{1}{|J|}\sum_{j\in J}\mathbb{E}\big[f(\boldsymbol{x})_j - f(\boldsymbol{x}_{J^c})_j\big]^2\right)^{1/2} \tag{26}$$

$$= \sigma\left(\frac{1}{|J|}\mathbb{E}\left\|f(\boldsymbol{x})_J - f(\boldsymbol{x}_{J^c})_J\right\|^2\right)^{1/2}. \tag{27}$$

Consequently, we can control Equation (1) as

$$\mathbb{E}_{x,y}\left\|f(\boldsymbol{x})-\boldsymbol{y}\right\|^2 + \mathbb{E}_{x,y}\left\|\boldsymbol{x}-\boldsymbol{y}\right\|^2 = \mathbb{E}_x\left\|f(\boldsymbol{x})-\boldsymbol{x}\right\|^2 + 2\,\mathbb{E}_{x,y}\langle f(\boldsymbol{x})-\boldsymbol{y},\boldsymbol{x}-\boldsymbol{y}\rangle \tag{28}$$

$$\leq \mathbb{E}_x\left\|f(\boldsymbol{x})-\boldsymbol{x}\right\|^2 + 2m\sigma\,\mathbb{E}_J\left(\frac{1}{|J|}\mathbb{E}\left\|f(\boldsymbol{x})_J - f(\boldsymbol{x}_{J^c})_J\right\|^2\right)^{1/2}. \tag{29}$$

This completes the proof of Theorem 2.

$\square$

## C  Dataset Constructions

**RGB Natural Images.** We construct the RGB natural image dataset from the ImageNet ILSVRC2012 Validation dataset that consists of 50,000 natural images. In particular, we follow [1] to generate noisy images by applying a combination of three types of noises to the clear images. The noises are Poisson noise ($\lambda = 30$), additive Gaussian noise ($\mu = 0, \sigma = 60$) and Bernoulli noise

($p = 0.2$). To be consistent to [1], we randomly crop 60,000 patches of size $128 \times 128$ from the first 20,000 images in ILSVRC2012 Val to construct the training dataset. Additional two sets of 1,000 images from ILSVRC2012 Val are used for validation and testing, respectively

**Hand-written Chinese Character Images.**   We generate the HànZì dataset with the code provided by [1]. The dataset is constructed with 13029 Chinese characters and consists of 78174 noisy images of size $64 \times 64$, where each noisy image is generated by applying Gaussian noise ($\sigma = 0.7$) and Bernoulli noise to a clear Chinese character image. Among the 78174 noisy images, 90% are used for training and validation, and the rest 10% are for testing.

**3D Fluorescence Microscopy Data.**   In order to show the capability of our approach to 3D images with inconsistent and untypical noise, we use the physically acquired 3D fluorescence microscopy data collected from Planaria (*Schmidtea mediterranea*) provided by [27]. The training data, consisting of 17005 3D patches of size $16 \times 64 \times 64$, is a mix of noisy images at three noise levels, collected under different conditions (C1, C2, and C3) of exposure time and laser power. The trained models are evaluated on 20 testing images of size $96 \times 1024 \times 1024$ at three different noise levels individually. From condition 1 (C1) to condition 3 (C3), the noise gets stronger, and input image quality gets worse.

**Grey-scale Natural Images.**   We follow [10] and use the same procedure as [4, 29, 22] to construct the BSD68 Dataset. For the training set, patches of size $180 \times 180$ are cropped from each of the 400 grey-scale version of images of range $[0, 255]$ from [15], where each image is treated with Gaussian noise ($\sigma = 25$). The trained models are then evaluated on 68 testing images first introduced by [20].

## D   Implementation Details

**Network Architecture.**   We use U-Net [19] as our neural network architecture and mainly follow [10] for the basic settings. To be specific, we apply a U-Net of depth 3 with convolutions of kernel size 3. The number of output feature maps from the initial convolution is set to 96. Batch normalization is applied by default unless further mentioned. Compared to [10], we remove the skip connection used in *Noise2Void* that add input to the network output, which contradicts the $\mathcal{J}$-invariant assumption in *Noise2Self* and prevents our model from learning from $\mathcal{L}_{inv}$.

**Mask Strategy and Training.**   By default, we randomly mask 0.5% of pixels for each training patch by saturated sampling and replace them with Gaussian noise ($\sigma = 0.2$). The scalar weight $\lambda_{inv}$ in the loss is set to 2 by default unless further mentioned. We apply data augmentations, including rotation and flipping, during the training for all datasets. For 3D images, the rotation is only applied on width and height dimension due to the anisotropy on depth. All the input images are normalized to satisfy the required condition in Theorem 1. We use learning rate decay during training, starting at 0.0004 and reducing the learning rate by 0.5 after each 5k iterations.

## E   Model Configurations

For the *convolutional blind-spot neural network*, we use the same network architecture and basic settings in [12]. For all the other deep learning-based methods, we fix the neural network (U-Net) architecture configurations and basic training settings for each dataset individually. The network and training configuration of our method basically follows the default settings in D. Exceptions are the scalar weights $\lambda_{inv}$ in the loss for BSD68. We adjust $\lambda_{inv}$ to 0.95 for the BSD68 Dataset according to the level of invariance error $\mathcal{L}$ during training. We apply the basic settings, such as mask sampling percentage, epoch numbers and batch sizes, of the two methods according to [10] for each dataset.

Besides, we skip the evaluation of some baseline methods on the Planaria dataset because they are not applicable. Among these methods, the BM3D algorithm does not apply to 3D images, and Noise2Noise is not applicable since no paired noisy image is available. Moreover, there is no public 3D version of the blind-spot network, which may need to deal with the anisotropy problem for the 3D images and the memory problem due to the additional branches required in a 3D setting.

Figure 5: **3D Microscopy Data**: Visualizations of testing results on the Planaria dataset. We compare the denoising quality among the traditional method *NLM*, the supervised method *CARE* [27], self-supervised baselines *Noise2Self* and our *Noise2Same*. From top to bottom, rows are reconstruction results from different noise levels (C1, C2 and C3 individually). From the left to the right, the columns are in the ascending order in terms of the denoising quality.

## F Denoising Results on the Planaria Dataset

The visualization of the denoising results on CARE (Planaria) is shown in Figure 5. The shown results are 2D projections from the 3D images.