[Reviews · NeurIPS 2020]

Review 1

Summary and Contributions: The paper argues that the denoising model trained through mask-based blind-spot approaches is not strictly J-invariant and minimizing the self-supervised loss with a J-invariant function is not optimal for self-supervised denoising. A regularization term that serves as a relaxed J-invariance measure of a model is derived from a self-supervised upper bound. The regularization term is then combined with a non-masked self-supervised reconstruction loss for training the model. Such a method can avoid learning identity mapping while keeping the center pixel value assessible during the self-supervised reconstruction training . This makes it different from existing methods which achieves strict J-invariance during training with blind-spot schemes or specific network designs. The proposed method also shows a little higher efficiency over the existing blind-spot-based approaches.

Strengths: 1. The proposed J-invariance regularization term allows implicitly avoiding learning identity mapping while keeping the center pixel value assessible in the model during self-supervised reconstruction training, which makes it different from existing approaches. 2. Theoretical analysis on the proposed regularization term is provided. 3. A simple yet intuitive self-supervised loss is proposed. The center pixel value is very useful for denoising. Blind-spot methods like N2V cannot assess the center pixel values, and thus the way they learn on the exploitation of the center pixel for denoising is not an optimal way. Specific network architectures with strict J-invariance cannot assess the center pixel value during both training and test, which is not optimal either and some post processing is needed. This paper provides a nice way to learn on dependency on the center pixel value for denoising, in which the center pixel value is directly assessible.

Weaknesses: With a double check on the paper after reading the rebuttal, I have the following concerns. 1. The major one is on correctness of the experimental comparison. I would like to share the results of the two most related blind-spot-based methods: N2S and Laine et al. [12], in their original implementations. The results tell a quite different story from what showed in the paper which used the versions with their own modifications. The original implementations of both N2S [1] and Laine [12] achieve much better results on BSD68, with more than 1.2dB PSNR, than that reported in the paper. Concretely, we use the model trained with the original authors' code and training scheme on BSD400, the same training dataset used in this paper. The test results on BSD68 are as follows: (1) Original N2S can achieve PSNR 28.12dB vs 26.98 reported in this paper with their own modifications. (2) Original Laine et al [12] can achieve PSNR 28.84dB vs. 27.15dB reported in the paper with their own modifications. In short, original implementations of N2S and [12] noticeably outperformed the proposed methods. There is a big gap between the results using original implementations and the one reported in the paper. In supplementary materials, It seems that the authors modified the original implementation (not sure, the description is not very clear). To be honest, I did not see good reasons why only include different results or the results from the modified implementations of the original papers, which are much lower than the results from their original implementations. *********************************************************************** 2. The claim in Section 3.1 is rather confusing. It says that "the denoising function f trained through mask-based blind-spot approaches is not strictly J-invariant, making Equation (2) not valid". Also, in the rebuttal, it says that "it is stated that the results in Table 1 indicate that the model f does not have the J-invariance, thus violating the assumption behind using the loss in Eqn. (3)." In N2S, during training, the denoising function f is J-invariant if we view f as the one equipped with masking. Thus, it does not violate Equation (2). Further, since Equation (2) is the loss for training which is not used in test, relating the J-invariance of a trained model in test to the conditions of Equation (2)(3) for training does not make sense.

Correctness: As described in the weakness, the claim in Section 3.1 is somehow misleading. The paper is trying to argue the importance of center pixel value in denoising. But there is confusion on the concepts of J-invariance/dependencies between training and test.

Clarity: The writing is fine to me. It is suggested that the details of model training/test/results of all compared methods, which are important to reproducibility, should be included in the main paper.

Relation to Prior Work: Adequate.

Reproducibility: Yes

Additional Feedback: I lowered the score due to the issues in the experimental comparison. Please see the weakness. The proposed method can be easily adapted to any existing self-supervised denoising. It is not clear why the author obtained the results of other methods by modifying the original implementation of them, instead of directly applying the proposed method with the same setting as these methods. In addition, there are issues in the claim in Section 3.1, which needs to be addressed.


Review 2

Summary and Contributions: After Rebuttal Summary: During the review phase, I did not realize that the PSNR scores of Laine et al and N2S reported in the paper are lower than what is reported in the original paper. I thank R1 for pointing this out. Given this new observation, I'm afraid that the conclusions in section 3.2 and hence the motivation of the paper doesn't hold anymore. However, I still think that characterizing the weak dependencies networks trained with masking has on the center pixel, and a new objective function that learns these dependencies systematically is interesting. The paper can be accepted if the inaccuracies in the results are fixed. I've updated my score to a weak accept. =========== The authors propose a new framework to train regular (non-J-invariant) neural networks for image denoising without ground truth. They do this via a new objective function that combines the MSE of the output of the network with the noisy image and another term which enforces that the network doesn't use the noisy pixel it is denoising. This loss function enables one to leverage the value of the noisy pixel we are denoising without collapsing the denoising function into identity.

Strengths: The proposed objective function is very straight forward and intuitive. The authors combine MSE of the network output with a noisy image with MSE of the network output of masked noisy image with again noisy image as target. The authors support this cost function by mathematically deriving that it's an upper bound of the supervised loss (eq 6). The proposed cost function works out of the box for different noise types, even when we don't have the knowledge of the distribution. To the best of my knowledge, the proposed work is novel. The work is relevant to the NeurIPS community and of practical importance for denoising.

Weaknesses: The proposed method is computally expensive during the training time, but it is a drawback suffered by all masking based methods. Further, knowing the noise model may give easy prior to incorporate in the framework, which I currently see no way of doing in this framework. Overall, in my opinion, the work doesn't have any significant limitations when comapred to the existing self supervised methods.

Correctness: Yes

Clarity: Yes, I enjoyed reading it.

Relation to Prior Work: Yes

Reproducibility: Yes

Additional Feedback:


Review 3

Summary and Contributions: According to the observation that J-invariant, which is required by most of the self-supervised denoising networks, leads to sub-optimal results, this paper proposes a novel loss for self-supervised image denoising without this constrain.

Strengths: 1. This paper is well-written, the logic is easy to follow. 2. The authors observe that J-invariant leads to sub-optimal results and propose a new loss to tackle it. 3. The proposed method does not need to know the noise model information.

Weaknesses: My only concern about this paper is the experimental results. Except for the psnr, I do not see a big visual difference relative to existing self-supervised trained methods. I recommend the authors provide some comparisons about visually perceptual metrics e.g. NIQE, BRISQUE. In addition, I think the authors need to provide some comparisons in real noisy dataset e.g. SIDD, NAM, DND. Since the noise model for real noisy images is unknown and it should be more suitable for self-supervised framework. minor issue: Why is the result of BM3D for BSD68 bolded in Table 3?

Correctness: yes

Clarity: yes

Relation to Prior Work: yes

Reproducibility: Yes

Additional Feedback: As pointed out by other reviewers, there are misstatements in the experiments in which the psnr of N2S in this paper is lower than in the original N2S paper. The authors need to address this issue.


Review 4

Summary and Contributions: This paper proposed a new method call noise2same for self-supervised image denoising. The proposed method is to optimize an upper bound of typical supervised loss, which contains a reconstruction loss and an invariance loss. Experimental results showed the effectiveness of the proposed method on synthetic noisy images.

Strengths: 1. The paper is well-organized and easy to follow. The idea of using the upper bound for self-supervised image denoising is very interesting. 2. There are lots of analysis and explanation for existing methods and the proposed method, which make the paper more convincing. 3. Consistently better results than the existing self-supervised methods in the experiments.

Weaknesses: 1. The idea of the paper is interesting but seems to be over-claimed. For example, in the abstract, it says the existing methods may be sub-optimal. But this is also true for the proposed method. The proposed method is derived by "an" upper bound of the typical supervised loss. It has not proven to the tightest bound. This is somehow misleading. In addition, the analysis in Sec. 4.2 did not show why the proposed method is better than the baselines. Most conclusions are obtained empirically. 2. The results are on synthetic noise. It will be great to apply the proposed method on real noisy image Benchmarking.

Correctness: Seems correct. Did not check the proof.

Clarity: satisfied.

Relation to Prior Work: yes

Reproducibility: Yes

Additional Feedback: Update after rebuttal: My score is lowered because some reviewer mentioned the inconsistency of baseline results. The results were misleading for readers.

[Author Response · NeurIPS 2020]

We thank the reviewers for their comments. The indices of references below are the same as references in our paper.

**Response to Reviewer #1**

*"The paper proposed a J-invariance measure as the loss term in self-supervised denoising."* **This summary of our work is neither accurate nor comprehensive. Our method should NOT be understood as just adding a $\mathcal{J}$-invariance measure term. (a)** The first term in Eqn.(6)/(8) is also different from the previous self-supervised (SS) loss in Eqn.(3). This brings benefits in efficiency, as shown in Fig.3; **(b)** We point out the practical results do not match the $\mathcal{J}$-invariance assumption in the theory behind using Eqn.(3). This is a flaw shared by previous SS denoising methods; **(c)** Our method is derived from a new and solid theory without the $\mathcal{J}$-invariance assumption, leading to improved performance.

*"The J-invariance is useful for overfitting ... Table I that relate D(f) and PSNR has little meaning."* **There is a misunderstanding about the results in Table 1. It is caused by statements in Line 115-117, which we will revise to make them accurate.** In fact, we are discussing the training process. $\mathcal{D}(f)$ is supposed to check whether $f$ has the $\mathcal{J}$-invariance, which is an intrinsic property of $f$. Although $\mathcal{D}(f)$ in Table 1 is computed on testing data, the same results ($\mathcal{D}(f) >> 0$) can be observed for when computed on training data, for any $f$ during training. **Therefore, results in Table 1 indicate that the model $f$ does not have the $\mathcal{J}$-invariance, thus violating the assumption behind using the loss in Eqn.(3) in training. Besides, we fix all model configurations and training settings except for the masking strategy to make Table 1 reasonable.**

*"Line 25 - Hard to agree N2N is supervised."* We follow prior studies [1,12] to categorize N2N as a supervised method.

*"Line 34,35: ... as they can still add AWGN to the noisy images to generate a noiser ones."* **It is not true.** [16,26] only work on additive and known noise models, so that the noise from the same distribution can be simulated. Since adding AWGN does not work with unknown noise models or noise types other than Gaussian, [16,26] may not be applicable. Furthermore, N2N requires the noise in the pairs of images to be independent and identically distributed. Adding AWGN to the noisy images changes the original noise distribution and dissatisfies the independence required by N2N.

*"Please explain how to determine the value of the weight of the J-invariance loss term."* In most cases, we follow Eqn.(6) to set the weight to its default value 2. However, when observing extremely imbalanced $L_{rec}$ and $L_{inv}$ during training, we adjust the weight to balance them, as described in Appendix E.

**Response to Reviewer #3**

*"Except for the psnr, I do not see a big visual difference relative to existing self-supervised trained methods."* The visual difference does exist and is sharp especially for the ImageNet dataset. We recommend zooming-in for a better view. We'll consider adding some zoomed-in views to the visualization to make it more clear.

*"I recommend the authors provide some comparisons about visually perceptual metrics e.g. NIQE, BRISQUE."* We follow prior denoising studies to use PSNR, in order to make consistent comparisons. NIQE and BRISQUE may not be suitable since they are not for evaluating the denoising performance and half of our datasets are not natural images.

*"In addition, I think the authors need to provide some comparisons in real noisy dataset ..."* The Planaria dataset is a real noisy dataset, on which our method still outperforms the baselines.

*"Why is the result of BM3D for BSD68 bolded in Table 3"* It was bolded by mistake. We will unbold it. In Table 3, the bolded values correspond to the best results among self-supervised deep learning methods.

**Response to Reviewer #4**

*"The idea of the paper is interesting but seems to be over-claimed ... This is somehow misleading."* We agree that the "sub-optimal" statement is inappropriate and will revise accordingly. Nevertheless, we provide convincing analytical and experimental results to show why and how Noise2Same outperforms the baselines.

*"In addition, the analysis in Sec. 4.2 did not show why the proposed method is better than the baselines ..."* **Sec 4.2 theoretically analyzes the invariance term. The analytical result suggests that the invariance term has the similar effect as the post-processing in previous methods, as discussed in Line 219-224. This explains why our method achieves better performance, especially in the case where post-processing is not applicable.** In addition, as we point out in Sec 3, the $\mathcal{J}$-invariance assumption is violated in practice, making the theory behind using the loss in Eqn.(3) not applicable. This potentially limits the performance of baselines. On the contrary, our method is derived from a new and valid theory. In this case, better performance is expected.

*"The results are on synthetic noise. It will be great to apply the proposed method on real noisy image Benchmarking."* The Planaria dataset is a real noisy dataset, on which our method still outperforms the baselines.

[Meta-Review · NeurIPS 2020]

"** What happened in the review phases: In the initial reviews, 3 out of of 4 reviewers (R2, R3, and R4) recommended acceptance, with R2 and R3 in particular judging the paper very positively (Top 50% of accepted NeurIPS papers). R4 was more mildly positive, bothered by felt overclaiming from a suggestion that the method be less sub-optimal than ohers, but still finding the idea of the paper worthy. R1 was more critical, in particular pointing out "confusion on the concepts of J-invariance/dependencies between training and test". Considering the author's response, the AC judges that the points, questions and criticisms raised in each initial review were overall well addressed (e.g. committing to revise the inappropriate “sub-optimal” statement pointed by R4). Except for one important point of contention that remained: authors tried to clarify the misunderstanding of R1 regarding J-invariance/dependencies in training and test connected to results in Table 1. This was brought up in the discussion phase between R1 and R2. R1 expressed that the author response had not clarified the issue, which severely undermined the motivation of their work. R2 shared his own understanding of the matter, which led to clearing the confusion and to R1 understanding the point the authors were making in the paper. ** AC REQUEST 1**: Based on this discussion, the AC concludes that the root of this confusion could be easily corrected if the paper carefully and explicity distinguished that mask-based blind-spot is used during training but not during use/testing (for Table 1, and the N2S model). Similarly (pointed out by R1) the sentence ""the denoising function f trained through mask-based blind-spot approaches is not strictly J-invariant, making Equation (2) not valid"" is confusing/incorrect, because the function thus-trained *if considered to include the masking, which we can also do at usage/test* is J-invariant and so respects Eq. 2. It ceases to be when used without the masking (but explain why do so?). This needs to be clarified. The AC however judges that it should be an easy fix. *********************************************** ** New concern raised during the discussion phase: ** In the discussion phase, R1 found that the results reported in the literature (confirmed by R1 rerunning original implementation) of N2S and Laine et al. [12] noticeably outperformed the proposed method, contrary to what the paper's results table reports. R2 checked the results reported in the literature and agreed. This perceived incorrect reporting of literature results is -- from the AC's reading of the situation -- what prompted all reviewers to significantly lower their initial scores. Reviewers updated their reviews accordingly. Once these updates were made available to the authors, they contacted the AC through CMT with clarifications on this point (see authors' message copied below), explaining that the reported results are results without the post-processing steps -- which is the fair comparison with the approach in the paper as it does does not suppose knowledge of a noise model (which the post-processing steps in the other methods use). Authors also point to several places in the paper where this is stated. As this rebuttal was received by the AC past the end of the discussion phase, he could not bring it up to the reviewers for their consideration. The AC however judges that the author's rebuttal of this concern is correct, and that the reported results are thus a fair comparison with the related methods from the literature. ** AC request 2: ** However the AC asks that the paper be updated by clarifying this explicitly in the caption of the results tables: The caption should report the performance obtained with post-processing informed by noise-model that appears in the literature, and clearly state why the table gives the performance without post-processing. ********************** ** AC's final judgment ** ********************** Based on the expressed reviews, the discussions, and his own reading, the AC judges that the paper contributes significantly to the field of self-supervised (image) denoising, (i.e. when the data is constituted only of noisy images, and the only assumption on the noise is that it be independent across dimensions and zero-mean). It does so by highlighting and showing limitations of prior related approaches, proposing a theoretically well-grounded novel approach, and experimentally showing its superiority (for the setting without assumption of a specific noise model). The AC judges that the new issues that appeared during the discussion phase and that caused the reviewers to significantly lower their scores are the result of misunderstandings, that can easily be avoided by simple fixes in the paper. The AC thus recommends acceptance, provided the simple clarifications described above (AC request 1 and AC request 2, as well as clarifications promised in the initial author response) are implemented in the revised version of the paper. Minor point: the AC also suggests to better *highlight* that an input normalization is applied for Theorem 1 (specifying that it is subtracting mean, dividing by stddev, as normalization can mean different things); and that it is *not applied* for Theorem 2, because it otherwise feels very strange that one has a sigma in the equation, while the other has none. *************************************** Message sent by the authors to the AC through CMT after the end of the discussion phase, after seeing updated reviews: Dear Area Chair, In the reviewer feedback, the four reviewers lower their scores from 5-8-8-6 to 3-6-6-5. The only reason is the concern about the baseline performance raised by Reviewer 1. However, we argue that the concern is invalid and misleading. We exactly follow the experimental settings in the original papers except for excluding the post-processing in [1, 12]. The key reason that different PSNRs are produced by Reviewer 1 is whether the post-processing is used. As we state in Section 5.1, we do not include the post-processing step in [1, 12]. That is because using the post-processing in [1, 12] requires specifying the noise type (from Gaussian, Poisson, and impulse) [12] or the variance of the noise [1], which is unavailable under our problem setting. We clearly explain this difference in our paper as quoted. Lines 93-99, Page 3: “In practice, it is common to have unknown noise models, inconsistent noises, or combined noises with different types, where the Bayesian post-processing is no longer applicable. In contrast, our proposed Noise2Same can make use of the entire input image without any post-processing. Most importantly, Noise2Same does not require the noise model to be known and thus can be used in a much wider range of denoising applications.” Lines 243-246, Page 7: “Note that ImageNet and HànZì have combined noises and Planaria has unknown noise models. As a result, the post-processing steps in Noise2Self [1] and the convolutional blind-spot neural network [12] are not applicable, as explained in Section 2”, Lines 254-256, Page 7: “On the other hand, the convolutional blind-spot neural network [12] suffers from significant performance losses without the Bayesian post-processing, which requires information about the noise models that are unknown.” "